



# 1  Impacts of the seasonal distribution of rainfall on vegetation productivity

# 2  across the Sahel

Wenmin Zhang[a,b], Martin Brandt[b], Xiaoye Tong[b], Qingjiu Tian[a*], Rasmus Fensholt[b]
[a] *International Institute for Earth System Sciences, Nanjing University, 210023 Nanjing, China*
[b] *Department of Geosciences and Natural Resource Management, University of Copenhagen, 1350 Copenhagen,*
*Denmark*
**Abstract**  Climate change in drylands has caused alterations in the seasonal distribution of rainfall including increased heavy
rainfall events, longer dry spells, and a shifted timing of the wet season. Yet, the aboveground net primary productivity (ANPP)
in drylands is usually explained by annual rainfall sums, disregarding the influence of the seasonal distribution of rainfall. This
study tests the importance of seasonal rainfall metrics (onset and cessation of the wet season, number of rainy days, rainfall
intensity, number of consecutive dry days and heavy rainfall events) on growing season ANPP. We focus on the Sahel and
north-Sudanian region (100-800 mm year$^{-1}$) and apply daily satellite based rainfall estimates (RFE-2.0) and growing season
integrated NDVI (MODIS) as a proxy for ANNP over the study period 2001-2015. Growing season ANPP in the arid zone
(100-300 mm year$^{-1}$) was found to be rather insensitive to variations in the seasonal rainfall metrics, whereas vegetation in the
semi-arid zone (300-700 mm year$^{-1}$) was significantly impacted by most metrics, especially by the number of rainy days and
timing (start and cessation) of the wet season. We analyzed critical breakpoints for all metrics, showing that growing season
ANPP is particularly negatively impacted after >10 consecutive dry days and that a rainfall intensity of 7 mm day$^{-1}$ is detected
for optimum growing season ANPP. We conclude that number of rainy days and the timing of the wet season are seasonal
rainfall metrics being decisive for favorable vegetation growth in semi-arid Sahel which needs to be considered when
modelling primary productivity from rainfall in the dryland's of Sahel and elsewhere.

* Corresponding author at: Kunshan Building Xianlin avenue,163#, Nanjing, China

  E-mail address: tianqj@nju.edu.cn (Q.Tian).



**Keywords:** Drylands, Seasonal rainfall metrics, Aboveground net primary production, Remote sensing

## Introduction

Livelihoods in most drylands depend heavily on aboveground net primary production (ANPP) in the form of
rain-fed crops and fodder for livestock (Abdi et al., 2014; Leisinger, KM, Schmitt, 1995). Annual ANPP thus plays
a decisive role in the context of livelihood strategies, food security and people's general wellbeing. ANPP in
drylands is primarily controlled by water availability with annual rainfall typically being limited to a short and
erratic wet season which can be highly variable between years. The current study focus on the sub-Saharan Sahel
zone, being one of the largest dryland areas in the world, which has been referred to as the region showing the
largest rainfall anomalies worldwide during the last century (Nicholson, 2000). Throughout the centuries, the
Sahelian population has adapted to this high rainfall variability and thus great inter-annual differences in available
ANPP can be balanced for example by a temporary abandonment of agriculture or seasonal livestock migration
(Romankiewicz et al., 2016). However, 21st century climate change threatens established coping strategies not only
by increasing inter-annual variability of rainfall regimes as a whole (Field, 2012; Kharin et al., 2007), but also by
an increasingly unpredictable seasonality and an altered number of heavy rainfall and drought events (Fischer et
al., 2013; Smith, 2011; Taylor et al., 2017). Improved knowledge on the vegetation response to the seasonal
variability of rainfall is thus crucial to better interpret the consequences of climate predictions of an altered global
hydrological cycle and to implement appropriate adaptation measures to climate change and food security in arid
and semi-arid lands like the Sahel.
While it is well known that the productivity of dryland vegetation is highly prone to variations in the
availability of water resources at the annual scale (Fensholt et al., 2013; Fensholt and Rasmussen, 2011; Herrmann
et al., 2005; Huber et al., 2011), there is a current lack of understanding how the seasonal distribution of rainfall
impacts on growing season ANPP (Rishmawi et al., 2016). Several studies have demonstrated the vegetation





sensitivity to the timing and magnitude of rainfall events based on individual plot data and model estimates (Bates
et al., 2006; Fay et al., 2000; Guan et al., 2014; Thomey et al., 2011), but the impact of specific rainfall metrics
(such as wet season length/timing, number of rainy days, rainfall intensity, number of consecutive dry days and
extreme events) on dryland vegetation productivity has rarely been studied at the regional scale potentially
including different biotic and abiotic controls. A few studies show that there is a strong dependency of Sahelian
vegetation growth on the timing of the start of the wet season (Diouf et al., 2016) and the frequency and distribution
of dry spells (Proud and Rasmussen, 2011), but currently no regional scale study assessing a variety of rainfall
metrics exists systematically analyzing the importance of rainfall metrics on vegetation growth as a function of
mean annual rainfall. Recent studies suggest that not only a shift in the timing of the wet season but also increasing
extreme events occur in Sahel (Panthou et al., 2014; Sanogo et al., 2015; Taylor et al., 2017; Zhang et al., 2017),
showing the need for a comprehensive understanding on how rainfall seasonality impacts on vegetation production.
The assessment of rainfall metrics capturing the seasonal variability and the associated impact on growing
season ANPP requires daily rainfall records and a robust methodology being able to extract the timing (start and
end) and duration of the wet season (Dunning et al., 2016; Liebmann et al., 2012). The availability and quality of
such data, and uncertainties in methods to extract the rainfall seasonality (Fitzpatrick et al., 2015) have complicated
regionally scaled studies on this topic. However, a new generation of high spatial resolution satellite based daily
rainfall estimates blended with station data has recently opened up the possibility to fill the gap between plot and
model based studies. Here our study aims at applying daily rainfall estimates to analyze and understand the impact
of seasonal rainfall metrics on vegetation productivity for the entire Sahel.

## Materials and methods

An empirical analysis is conducted based on gridded information of rainfall metrics based on daily satellite
estimates and seasonally integrated NDVI (hereafter NDVI SIN) as a proxy for the growing season ANPP. The





period of analysis covers 2001-2015 which allows for a per-pixel analysis including state of the art Earth
observation datasets of both African Rainfall Estimation (RFE-2.0) and Moderate Resolution Imaging
Spectroradiometer (MODIS) vegetation. The rainfall metrics include: number of rainy days, daily intensity, heavy
rainfall events, number of consecutive dry days and seasonal rainfall amount which were analyzed as explanatory
variables for the observed spatial variability in seasonal vegetation productivity along the gradient of mean seasonal
rainfall.

*Study area*
The Sahelian zone covers arid and semi-arid biomes and is one of the world's largest dryland areas bordering
the Sahara Desert to the north (Fig. 1). The delineation of the Sahel is often done by using average annual rainfall
isohyets, with the northern boundary at 100 mm year$^{-1}$ and the southern boundary defined by 700 mm year$^{-1}$ (Lebel
and Ali, 2009). For this study we expanded the southern limit towards the Sudanian zone until 800 mm annual
rainfall to include also the zone where rainfall as the primary climatic forcing variable on vegetation productivity
is expected to level off (Fensholt et al., 2013; Huber et al., 2011; Kaspersen et al., 2011). The Sahel is characterized
by a unimodal rainfall regime and the landscape is generally flat and consists of large plains interspersed with sand
dunes and rocky formations. The large stretches of plains are mainly used for grazing and subsistence cultivation.
The northern parts of the Sahel are dominated by open and sparse grass- and shrublands, while cropland, open
woody vegetation and deciduous shrublands characterize the southern parts (Breman and Kessler, 1995).

*RFE rainfall data*
RFE-2.0 is a gauge-satellite blended rainfall product developed by the NOAA Climate Prediction Center (CPC)
(Herman et al., 1997) gridded at $11 \times 11$ km spatial resolution and widely used to explain changes in vegetation
productivity (Fensholt and Rasmussen, 2011). RFE-2.0 uses additional techniques to better estimate precipitation



while continuing the use of cloud top temperature and station rainfall data, in all four input data sources are included:
(1) daily Global Telecommunications System (GTS) rain-gauge data, (2) Advanced Microwave Sounding Unit
(AMSU)-based rainfall estimates, (3) Special Sensor Microwave Imager (SSM/I)-based estimates, (4) the
Geostationary Operational Environmental Satellite (GOES) precipitation index (GPI) calculated from cloud-top
infrared (IR) temperatures on a half-hourly basis.

*Deriving rainfall seasonality metrics*
The method used to identify the onset and cessation of the wet season is referred to Liebmann et al. (2012)
and applicable to multiple datasets for precipitation seasonality analysis across the African continent (Dunning et
al., 2016). As is described in Liebmann et al. (2012), the climatology wet season is initially determined by the
climatological cumulative daily rainfall anomaly, *A(d)*, calculated from the long-term (2001-2015) average
rainfall ($R_i$) for each day of the calendar year minus the long-term annual-mean daily average ($\bar{R}$). The day with
minimum value ($d_s$) is defined as the start of the wet season and the maximum point ($d_c$) marks the end of the wet
season.

105    (1)

$$A(d) = \sum_{i=1\,Jan}^{d} R_i - \bar{R}$$

Subsequently, the onset and cessation were calculated individually for each year and each grid point. For each
year the extraction of the seasonality of the wet season was based on equation (2). The daily cumulative rainfall
anomaly *A(D)* on day ($P_i$) was computed for each day in the range $d_s$ -50 to $d_c$ +50 for each year and the day with
minimum value is considered as the onset of the wet season.

111    (2)



$$A(D) = \sum_{j=d_s-50}^{D} P_i - \bar{R}$$

Once the onset and cessation dates of the wet season for each year were found, the remaining variables were
calculated (Table 1). Fig. 2a illustrates an example of daily rainfall for the grid point (13.5° N, 5.0° W) in 2001 and
the corresponding cumulative daily anomaly curves are shown in Fig 2b. The blue and red lines signify *A(d)* and
*A(D),* respectively. The range of minimum and maximum points in the blue line denotes the climatological wet
season (Liebmann et al., 2012). The wet season of each individual year was then determined based on the daily
precipitation observations covered by the climatological wet season. Areas where the annual minimum occurs
after the 1st of October (desert areas) were excluded from further analysis (Diaconescu et al., 2015).

*Estimation of growing season ANPP*
We used the NDVI SIN derived from the MODIS/Terra surface reflectance product (MOD09Q1 collection 6)
as a proxy for the growing season ANPP. NDVI was calculated from the MODIS red and near-infra red bands (8
day composites) for the period 2001-2015. The NDVI SIN was derived using TIMESAT (Jönsson and Eklundh,
2004), a widely used tool to extract vegetation seasonal metrics. For this study, we applied the Savitzky-Golay
filter and determined the start and end of season at 20% and 50% of the amplitude respectively. The method is well
established and proven to be a reliable proxy for the growing season ANPP in Sahel (Fensholt et al., 2013; Olsson
et al., 2005). NDVI data was aggregated to the resolution of RFE-2.0 (11×11 km) using a bilinear resampling
method. Both Globeland30 and ESA CCI (2010) land cover maps were used to mask water bodies, irrigated and
flooded areas.



### *Statistical analyses*


The Pearson's correlation coefficient was used to measure the relationship between growing season ANPP
and rainfall metrics. Additionally, Generalized Additive Models (GAMs) (Wood, 2017) implemented in the R
computing environment (Team, 2014) were applied to derive smooth response curves with seasonal rainfall amount
as the explanatory variable and the linear coefficients (averaged over 10 mm rainfall steps) as the response variable.
The models were parameterized assuming normal error distributions. Furthermore, a random forest ensemble
learning method (Breiman, 2001) was used to analyze the relative importance of individual rainfall variables on
growing season ANPP as a function of the seasonal rainfall amount. This algorithm produces multiple decision
trees based on bootstrapped samples and the nodes of each tree are built up by an iterative process of choosing and
splitting nodes to achieve maximum variance reduction. Thus the variables with highest difference are considered
as most important factors. All pixels based on 15-year averages of seasonal rainfall metrics and ANPP were used
for this analysis. Additionally, a multiple regression analysis was applied to identify and map the spatial distribution
of the relative importance of the three most important seasonal rainfall metrics (onset and end of the wet season
and rainy days) explaining the growing season ANPP at the per-pixel level (based on a 15-year time series).
A piecewise regression was used to identify breakpoints (Muggeo, 2003), i.e. critical thresholds in the
relationship between rainfall metrics and vegetation growth. A breakpoint is an indication that the vegetation
response to changes in a given rainfall metric surpasses a threshold beyond which vegetation functioning is
significantly altered. Such a threshold, at the level of individual rainfall seasonality metric, provides an indication
of rainfall conditions beyond which vegetation does not tolerate further stress without a marked impact on the
growing season ANPP. The 95th percentile of NDVI SIN was selected to represent the potential vegetation
productivity attainable for a given rainfall amount (Donohue et al., 2013) and the breakpoint regression was applied
to the potential vegetation productivity and corresponding rainfall seasonal metrics.





## Results

### *Spatial pattern of rainfall metrics*

A clear north-south gradient was observed for the onset, rainy days (RD), heavy rainfall events (R95p) and consecutive dry days (CDD) based on a 15-year (2001-2015) average value, with a later onset, more rainy days and less heavy rainfall events towards the south (Fig.3a,c,e). The cessation of the wet season (Fig.3b) shows some longitudinal differences with the latest dates found in the eastern Sahel, followed by the western Sahel and the central Sahel showing the earliest cessation dates. A considerable difference between southeastern and southwestern Sahel (Fig. 3d) was observed in the rainfall intensity (SDII). The relatively low rainfall intensity in the southeastern Sahel is mirrored by considerably higher numbers of rainy days.

### *General response of growing season ANPP to rainfall metrics*

The median growing season ANPP clearly follows the mean seasonal rainfall gradient (Fig.4) with a near linear relationship with other biotic and abiotic factors (e.g. soil texture, nutrients, species composition, fire regime and seasonal rainfall distribution) causing a wide range of values as indicated by the different quantile values.

The relationships between all the rainfall metrics and growing season ANPP were found to be significant ($p<0.001$) and the correlation coefficient ($r$) varied between -0.78 and 0.82 (Fig.5). For the region as a whole, the number of rainy days was identified as the most important metric impacting on the growing season ANPP ($r=0.82$) (Fig.5c), closely followed by heavy rainfall events (R95p) ($r=-0.78$) (Fig.5e), cessation ($r=0.69$) (Fig.5b) and onset ($r=0.61$) (Fig.5a) of the wet season. The impact of consecutive dry days (CDD) on growing season ANPP was also relatively strong ($r=-0.60$) (Fig.5f), whereas a rather weak correlation was observed between growing season ANPP and rainfall intensity (SDII) ($r=0.28$) (Fig.5d). For all rainfall metrics except R95p and CCD, the plots show a



bimodal distribution of the points, which is caused by the differences in the east-west patterns of the spatial
distribution of rainfall metrics (onset, cessation, RD and SDII) in Sahel as reported in Fig.3.

*Response of growing season ANPP to rainfall metrics along the rainfall gradient*

181          Although the relationship between growing season ANPP and seasonal rainfall amount (Pt) obviously changes

along the rainfall gradient (Fig. 4), variations in seasonal rainfall distribution can cause considerable changes in
growing season ANPP under the same rainfall gradient (Fig. 5). The influence of seasonal rainfall distribution on
growing season ANPP was thus analyzed under different mean annual rainfall values along the north (low rainfall)
to south (high rainfall) gradient (Fig. 6). The dependency of growing season ANPP on rainfall metrics is clearly
seen to be a function of the seasonal rainfall amount. Below ~300 mm year$^{-1}$, the vegetation seems to be stable and
rather insensitive to variations of the rainfall metrics, however, above 300 mm year$^{-1}$, the impacts can be clearly
seen by a larger spread in growing season ANPP values for a given amount of seasonal rainfall (Fig. 6). For example,
the growing season ANPP decreases strongly with a later onset, an earlier cessation of the wet season, a smaller
number of rainy days, a higher rainfall intensity, more heavy rainfall events and longer dry spells. Above ~700 mm
year$^{-1}$, the vegetation shows again a reduced sensitivity to variations of the wet season by a convergence of growing
season ANPP values irrespective of the seasonal rainfall metric value.

194          The Pearson's correlation coefficient was used to quantify the strength of the impact of individual seasonal

rainfall metrics on growing season ANPP as a function of seasonal rainfall (Fig. 7). In general, the impacts of
rainfall metrics on vegetation are distinctive along the 100-800 mm year$^{-1}$ gradient with a peak in $r$ values
(respectively positive or negative dependent on the rainfall metric) for most metrics around 700 mm year$^{-1}$ followed
by a sharp drop-off (Fig. 7b, c, d, e) (cessation of wet season, RD, SDII and R95p). It should be noted that modelling
uncertainties increase for all rainfall metrics due to fewer observations in the lowermost seasonal rainfall total bins.



For the wet season the pattern of onset is reversed with a sharp increase in *r* values from 700-800 mm year⁻¹. The
number of consecutive dry days (CDD) shows moderate *r* values balancing around zero generally indicating less
importance of the CDD variable on growing season ANPP along the rainfall gradient analyzed.


205        The relative importance of the individual rainfall metrics on growing season ANPP was assessed based on a

random forest model (Fig. 8a). The explained variance of growing season ANPP explained by rainfall metrics (blue
line in Fig. 8a) is increasing with mean annual rainfall up to 600-700 mm year⁻¹ from where the degree of explained
variance decreases, which corresponds with the results presented in Fig. 4 (the widest belt of the quantile values is
observed for rainfall of 600-700 mm year⁻¹, suggesting that the seasonal rainfall metrics additional to the seasonal
rainfall amount is increasingly important in this rainfall zone) and in Fig. 7. The cessation of the wet season was
identified as the most important factor controlling growing season ANPP in semi-arid areas of Sahel (300-700 mm
year⁻¹), followed by the onset and number of rainy days. As measured by the relative importance, these three rainfall
metrics together accounted for 60-70% of the variance explained by all rainfall metrics for all seasonal rainfall
amounts. In arid areas (100-300 mm year⁻¹) the number of rainy days was found to be the most important variable.

216        The spatial distribution of the relative importance of the onset and cessation of the wet season and number of

rainy days on growing season ANPP was identified at the pixel-level for 2001 to 2015 (Fig. 8b).  At Sahel scale,
the number of rainy days (bluish colors) is observed to be the dominating factor, followed by the onset of the wet
season (reddish colors) and cessation (greenish colors). There are no clear signs of a latitudinal or longitudinal
dependency on which rainfall metric is dominating, but some clustering is evident with a predominance of rainy
days influence in the Western Sahel with limited influence by the cessation date on the growing season ANPP.




*Critical points for growing season ANPP*

A piecewise regression between growing season ANPP and seasonal rainfall metrics was applied to identify if critical breakpoints in the relationship between growing season ANPP and rainfall metrics exist (Fig. 9). We found that the most evident thresholds (average values for the Sahel zone) in relation to seasonal rainfall metrics influence on growing season ANPP relate to the onset of the wet season (Fig. 9a), the rainfall intensity (SDII) (Fig. 9d) and consecutive dry days (CDD) (Fig. 9f). If the onset of the wet season is later than day of year 134 this will have an increasingly negative effect on growing season ANPP as a function of the onset delay and contrastingly if the onset starts earlier than day of year 134 this will also have an increasingly adverse effect on growing season ANPP. Also, an optimum of rainfall intensity of 7 mm day$^{-1}$ was detected; if rainfall intensity exceeds 7 mm day$^{-1}$, vegetation productivity starts to be negatively affected, whereas a lower intensity will also negatively impact on growing season ANPP. There was a pronounced decline in growing season ANPP as a function of number of consecutive dry days. However, when CDD exceeds 10 days a breakpoint in the curve is detected, as dry spells of this magnitude leads to a strong reduction in vegetation growth. The number of rainy days (Fig. 9c) was linearly related to growing season ANPP until 75 days, beyond which RD becomes less decisive for the amount of growing season ANPP. Similarly, heavy rainfall events (fraction of annual rainfall events exceeding the 95th percentile) (Fig. 9e) were negatively related to growing season ANPP, until a certain threshold (larger than 0.83), from where the rather low amount of vegetation loses sensitivity to even more extreme seasonal distributions. Finally, the cessation date of the wet season (Fig. 9b) was shown to be nearly linearly related to growing season ANPP. A breakpoint is detected by the piecewise regression algorithm around day 257, beyond which the delay in cessation date was slightly more favorable for higher growing season ANPP yields as compared to a cessation date before day 257.



## Discussion

This study presents first empirical evidence on the impact of rainfall seasonality on vegetation productivity at
regional scale and the results provide a clear picture on the importance of six rainfall metrics on growing season
ANPP under different rainfall conditions (i.e. mean annual rainfall). The Sahel zone is often defined from isohyets
of annual rainfall as a common denominator of the hydrological conditions of the region. It was however shown
here, that considerable east-west differences occur in several of the seasonal rainfall metrics analyzed with a much
higher number of rainy days and corresponding lower rainfall intensity in the southeastern Sahel as compared to
the southwestern part. Variability in the rainfall intensity is shown here to influence the growing season ANPP
generated over a growing season (Fig. 9d) and hence spatio-temporal changes in rainfall intensity (but characterized
by the same amount of seasonal rainfall) will impact vegetation productivity. This has important implications for
the use of the rain use efficiency (RUE) (Houerou, 1984) or RESTREND approach (Evans and Geerken, 2004)
which is derived from annually or seasonally summed rainfall and commonly used as an indicator for land
degradation (Archer, 2004; Bai et al., 2008; Fensholt and Rasmussen, 2011; Prince et al., 1998; Wessels et al.,
2007) as discussed in Ratzmann et al. (2016). Interestingly, a limited impact of rainfall seasonality on growing
season ANPP was found in arid lands below 300 mm year$^{-1}$ rainfall, suggesting that the species composition is
adapted to rainfall variation, and that the rather sparse vegetation cover is able to effectively utilize rainfall
independent of the seasonal distribution. This is very different in the semi-arid and northern sub-humid zone (300-
700 mm year$^{-1}$), where variations in rainfall seasonality are found to be more closely linked with variations in
growing season ANPP. This implies that a favorable distribution of rainfall may lead to increased productivity, as
it was observed in Senegal between 2006 and 2011 (Brandt et al., 2017), but below average rainfall conditions with
an unfavorable distribution lead to an immediate reduction in vegetation cover and growing season ANPP.
Not surprisingly, the number of rainy days showed the highest positive correlation with growing season ANPP
(Fig. 5c), with increasing productivity along with increasing rainy days, up to 75 days, where the relation weakens



(Fig. 9c). The importance of this metric is closely followed by the heavy rainfall events, which were negatively
correlated with growing season ANPP (Fig. 5e), and decreases the productivity until heavy rainfall events reach a
share of 80%, which leads to a constant level of low vegetation productivity (Fig. 9e). The importance of the timing
of the wet season, i.e. the start and cessation, increases rapidly along the rainfall gradient, having the highest impact
on growing season ANPP in the semi-arid zone (300-700 mm year$^{-1}$). The reason for the importance of the timing
of the onset of the growing season on the growing season ANPP (Fig. 9a) with an almost linear decrease in growing
season ANPP as a function of onset delay should be found in the predominance of annual grasses which are
photoperiodic (Vries and Djitèye, 1982). The end of season is thus controlled by day length and do therefore not
compensate for a delay in the onset. Also a too early onset was found to decrease growing season ANPP, which is
likely to be associated with so-called "false starts" of the growing season. Often, an early start of the growing
season is accompanied by a significant number of dry-days occurring shortly after the start of the rainy season with
a detrimental impact upon plant growth (Proud and Rasmussen, 2011). Finally, we found that the general impact
of consecutive dry days is difficult to quantify (Fig.7f). However, a rather clear critical threshold of 10 consecutive
days without rainfall was found to have an increased adverse impact on the growing season ANPP. This threshold
of 10 days as an average for Sahel is related to the depletion time of the upper layer soil water and the root depth
of the herbaceous stratum (primarily annual grasses) but will vary spatially with different soil and vegetation types
(Vries and Djitèye, 1982).
Several studies have reported a tendency towards an earlier start of the wet season (Sanogo et al., 2015; Zhang
et al., 2017), but projections predict a delay of the wet season in the later 21th century (Biasutti and Sobel, 2009;
Guan et al., 2015). Moreover, an increase in heavy rainfall events and prolonged dry spells are observed and
projected for the future (Sanogo et al., 2015; Taylor et al., 2017; Zhang et al., 2017). Our results show that the
semi-arid zone will be most prone to these projected changes, and an increase in heavy rainfall events, a delay in
the start of the wet season and dry spells exceeding 10 days will cause a significant reduction in vegetation
productivity, although the annual rainfall amount may be constant or even increasing. Disregarding the importance
and impact of varying rainfall distribution on vegetation productivity leads to a bias in any ANPP prediction, and
this knowledge should be implemented in any prediction and estimation of ANPP, both for ecosystem models and
remote sensing based analyses, especially in the context of food security. Although this study did not include a
temporal change component of the metrics analyzed, the length of high quality time series data of both vegetation
productivity and rainfall with a daily temporal resolution does provide the possibility for adding a temporal
dimension which will be pursued in a future study.

## Summary and conclusion

In this study we analyzed the impact of seasonal rainfall distribution (represented by onset and cessation of the
wet season, number of rainy days, rainfall intensity, number of consecutive dry days and heavy rainfall events) on
growing season ANPP for the Sahelian zone. Overall, a clear north-south gradient was observed for the onset, rainy
days, heavy rainfall events and consecutive dry days, but also considerable differences in cessation date, number
of rainy days and the rainfall intensity were observed between Eastern and Western Sahel. We found the strongest
relationship between growing season ANPP and the number of rainy days (r=0.82), closely followed by a negative
correlation between growing season ANPP and heavy rainfall events (r=-0.78). Growing season ANPP in the arid
zone (100-300 mm year$^{-1}$) was rather insensitive to variations in the rainfall metrics, whereas vegetation in the
semi-arid zone (300-700 mm year$^{-1}$) was significantly impacted by most metrics, especially by the number of rainy
days and timing (start and cessation) of the wet season. Finally, the critical breakpoints analysis between growing
season ANPP and all rainfall metrics show that the growing season ANPP are particularly negatively impacted
after >10 consecutive dry days and that a rainfall intensity of 7 mm day$^{-1}$ is detected for optimum growing season
ANPP. Overall, it can be concluded that seasonal rainfall distribution significantly influence ANPP and the effect
of different rainfall metrics is observed to vary along the north-south rainfall gradient. These findings have




important implications for the sheer amount of dryland studies in which annually or seasonally summed rainfall and ANPP are used to derive indicators of land degradation or anthropogenic influence (e.g. the use of RUE and RESTREND). When studying subtle changes in dryland vegetation productivity based on time series of satellite data, as caused by both climate and anthropogenic forcing, it is essential also to consider the potential effect from changes in the rainfall regime as expressed in the seasonal rainfall metrics studied here.

## Acknowledgements

This study is jointly supported by the European Union's Horizon 2020 research and innovation programme under the Marie Sklodowska-Curie grant agreement (project BICSA number 656564), China Scholarship Council (CSC, 201506190076), and the Danish Council for Independent Research (DFF) project: Greening of drylands: Towards understanding ecosystem functioning changes, drivers and impacts on livelihoods, and Chinese National Science and Technology Major Project (03-Y20A04-9001-15/16).

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




**Table 1** Rainfall metrics describing the seasonality and extreme events

| Index name | Definitions | Units |
|---|---|---|
| Onset of wet season (Onset) | The minimum value in the accumulative anomaly of daily rainfall | day of year |
| Cessation of wet season (Cessation) | The maximum value of the accumulative anomaly of daily rainfall | day of year |
| Rainy days (RD) | Number of days with rainfall >=1 mm between the onset and the cessation of the wet season | days |
| Rainfall intensity (SDII) | Ratio of annual total rainfall and number of rainy days ≥1 mm | mm day$^{-1}$ |
| Heavy rainfall events (R95p): | Fraction of annual rainfall events exceeding the (2001–2015) 95th percentile | % |
| Consecutive dry days (CDD): | Maximum number of consecutive days with rainfall <1 mm | days |
| Seasonal rainfall amount (Pt) | Rainfall amount during the wet season | mm year$^{-1}$ |





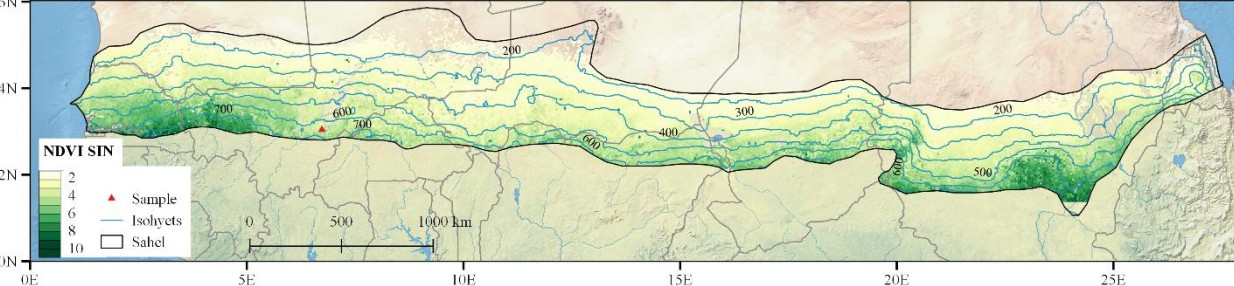


Fig 1. Study area outlining the Sahelian region (black color; 100-700 mm annual rainfall). NDVI SIN (seasonal integral) is
based on a 15-year (2001-2015) average using MODIS data; the red grid point (13.5° N, 5.0° W) is used to illustrate the
extractions of onset and cessation of the wet season shown in Fig. 2. Isohyets are based on a 15-year (2001-2015) average of
the seasonal rainfall amount (RFE-2.0).



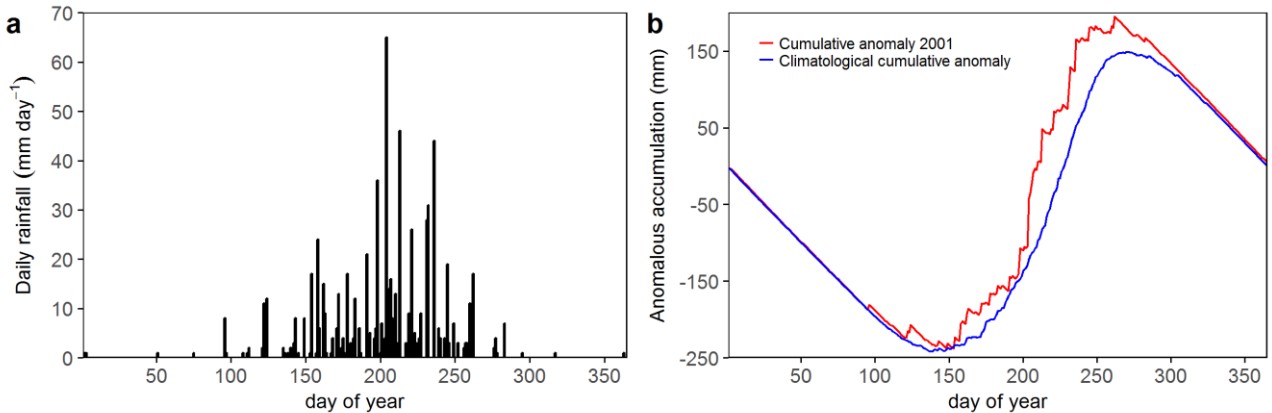


Fig.2. (a) Daily rainfall distribution and (b) anomalous accumulative curve for the grid point 13.5˚ N, 5.0˚ W (shown in Fig.

1) for the year of 2001. The blue line (accumulated anomaly) is computed from a 15-year (2001-2015) average of daily rainfall.


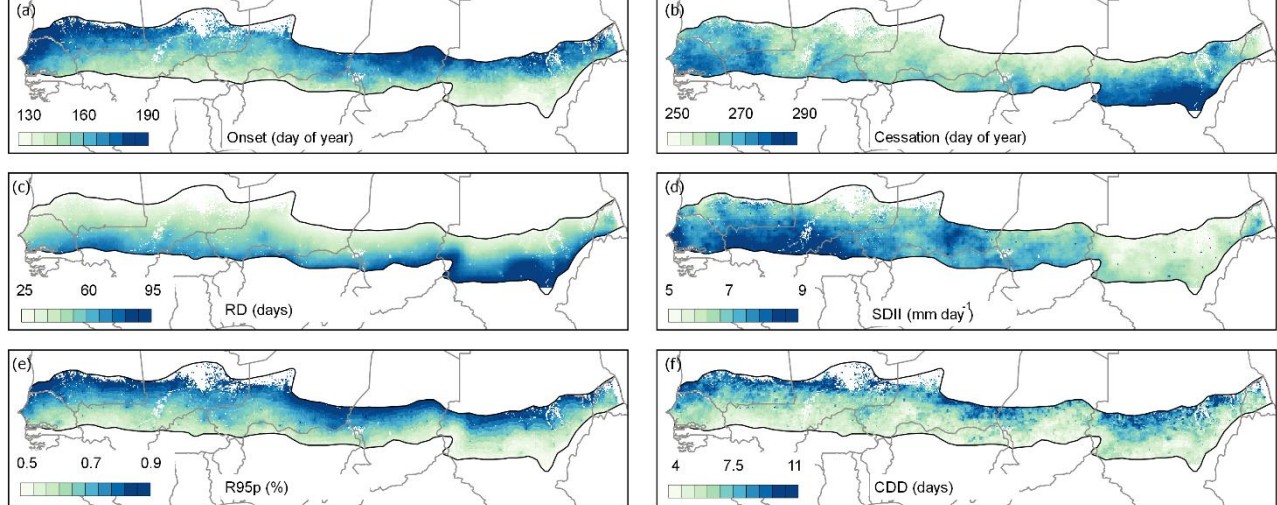

Fig.3. Spatial distribution of average seasonal rainfall metrics a) onset of wet season (day of year); b) cessation of wet season

(day of year); c) rainy days (days); d) daily intensity (mm day⁻¹); e) heavy rainfall events (%); f) consecutive dry days (days)

based on 15-year averages (2001-2015). Pixels within the study area are masked (white color) in accordance with the

description in the methods section.






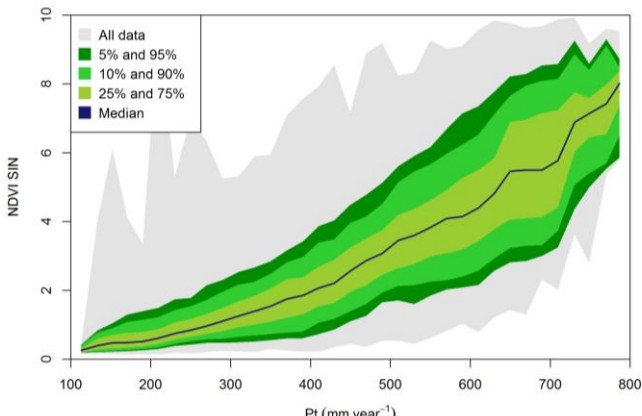


Fig.4. Growing season ANPP (NDVI SIN) as a function of mean seasonal rainfall amount (Pt) plotted as quantiles for the
Sahel-Sudanian zone (pixel averages for 2001-2015).


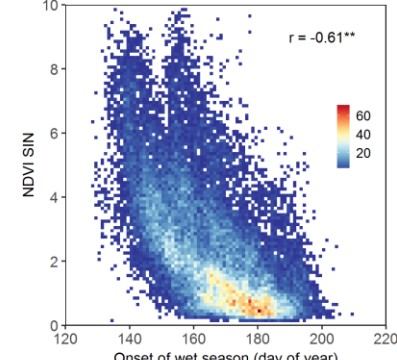
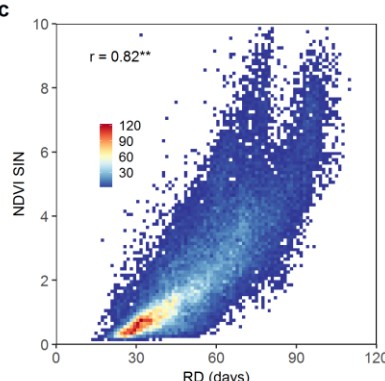






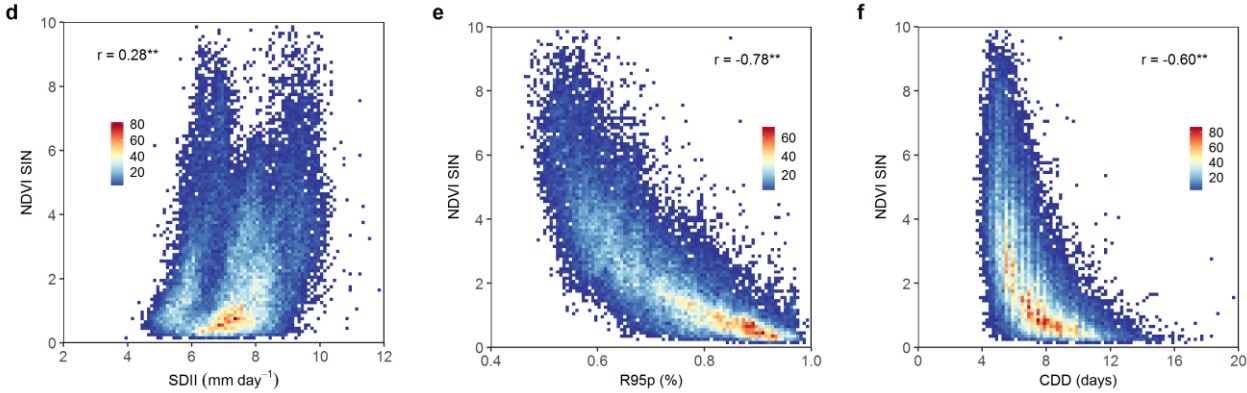


Fig.5. Density scatterplot showing the relationships between rainfall metrics and growing season ANPP (NDVI SIN). All
analyses are based on 15-year averages (2001-2015). ** denotes $p< 0.001$. All points (n=30862) are located between 100 and
800 mm annual rainfall.


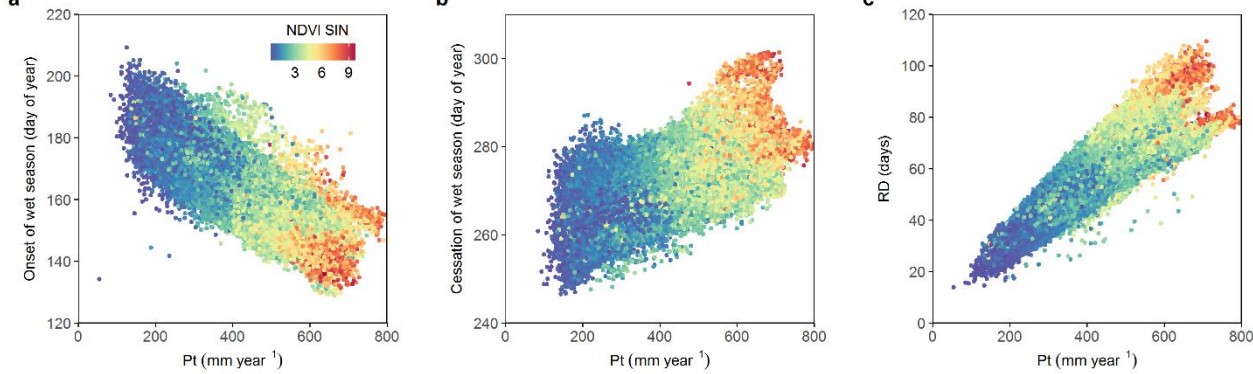




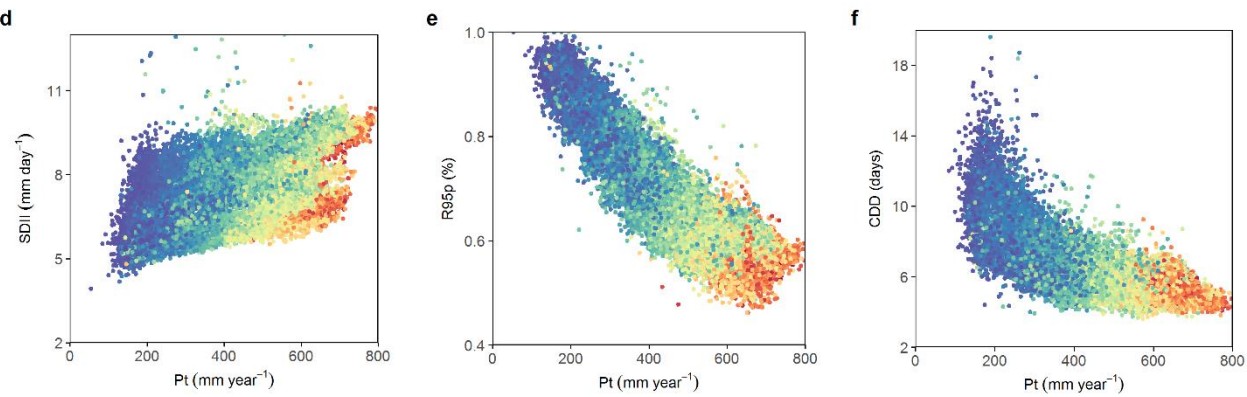

Fig. 6. Relationships between seasonal rainfall metrics and growing season ANPP as a function of seasonal rainfall amount based on 15-year averages (2001-2015).

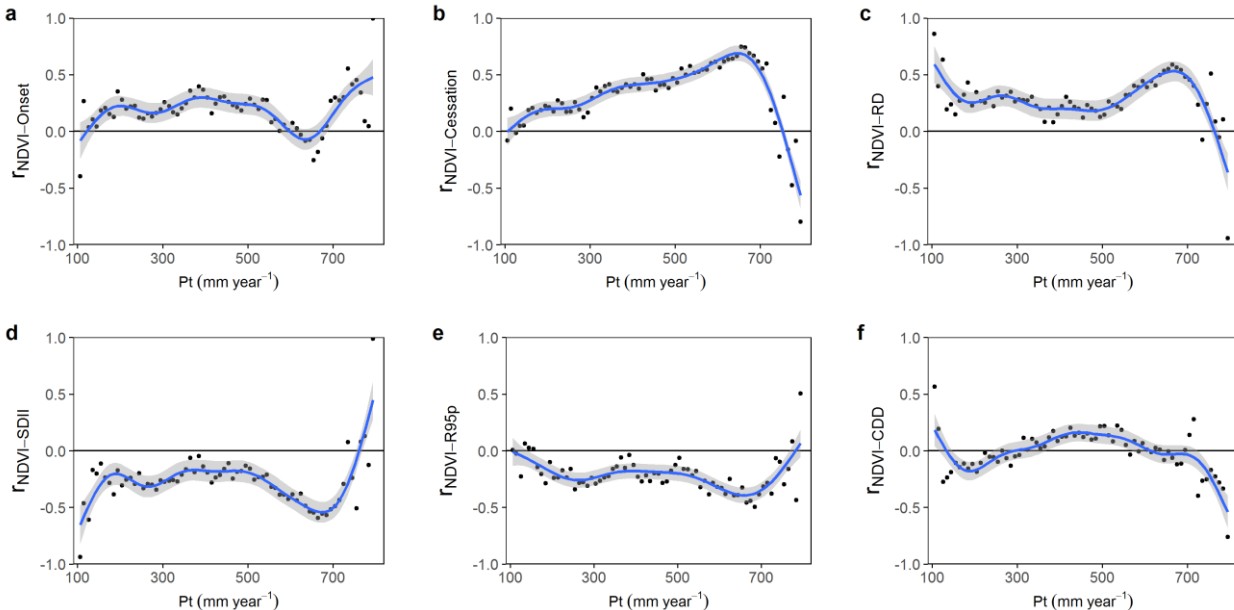

Fig.7. Effects of rainfall metrics on growing season ANPP as a function of seasonal rainfall amount. The Pearson's correlation between growing season ANPP and rainfall metrics are shown for each 10 mm interval. The lines are GAM fitting curves and shading represents the 95% confidence intervals of the fitting.






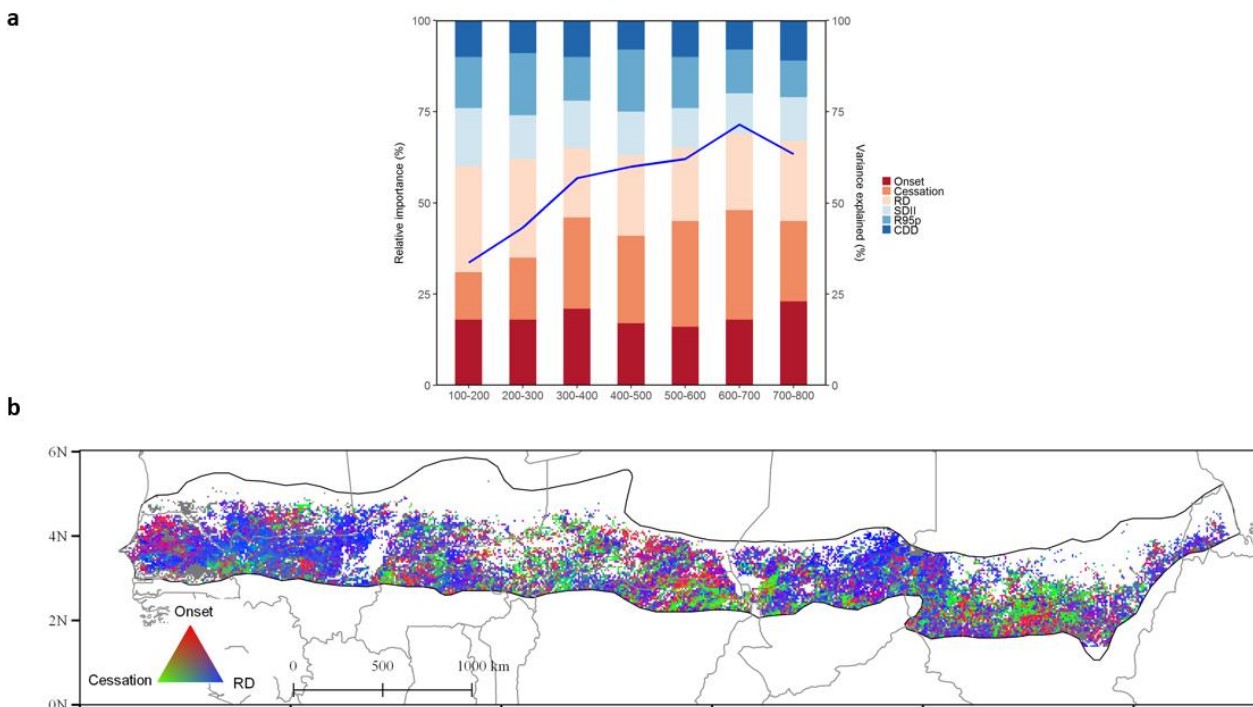


Fig.8. a) Generalized relationships between growing season ANPP and Onset, Cessation, RD, SDII, R95p, CDD as a function
of seasonal rainfall amount (100 mm intervals) based on 15-year average values (relative importance in %). The blue line
shows the overall variance of growing season ANPP explained by the rainfall metrics per 100 mm seasonal rainfall amount
based on the random forest method. b) Spatial distribution of the relative importance of onset, cessation of wet season and rainy
days to growing season ANPP for 2001-2015 based on a multiple regression. Pixels within the study area are masked (white
color) in accordance with the description in the methods section.





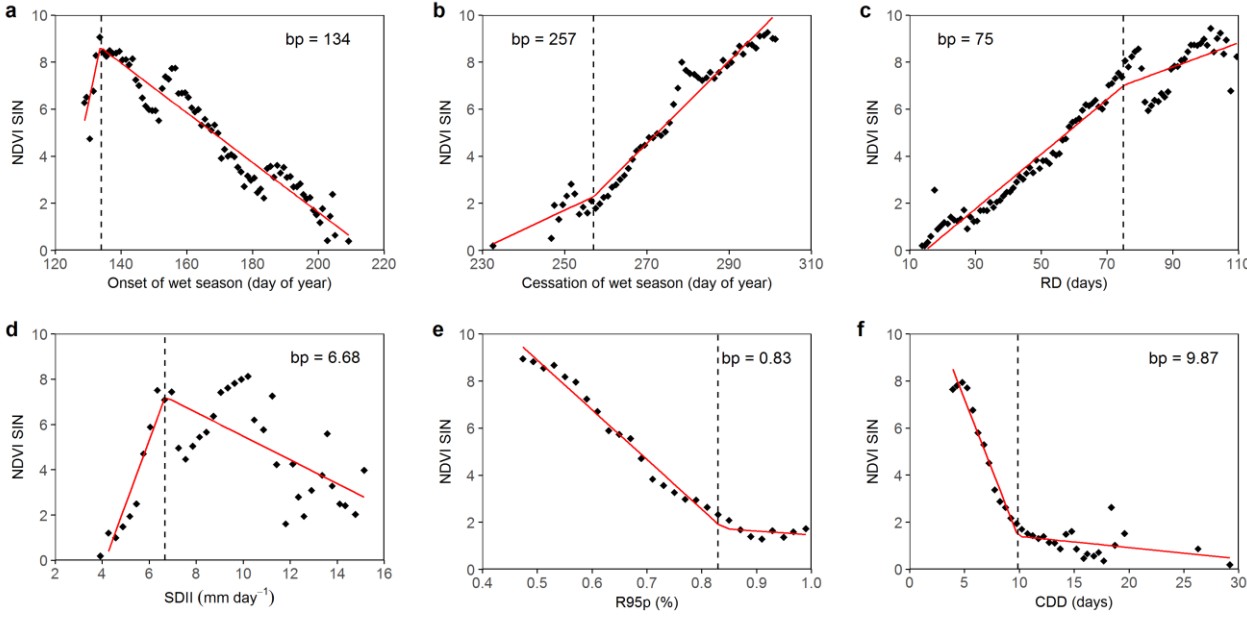


Fig. 9. Growing season ANPP (individual points represent 95th percentile of NDVI SIN value for each rainfall metric bins)

plotted against rainfall metrics. Solid red lines denote piecewise regression between growing season ANPP and rainfall metrics

and dashed lines indicate the breakpoint (bp) for rainfall individual variables.
