# Peer review of "Impacts of the seasonal distribution of rainfall on vegetation productivity"

_Biogeosciences, 2017_

## Referee Comment (RC1) · M. Marshall (Referee) · 29 Sep 2017

The manuscript titled, "Impacts of the seasonal distribution of rainfall on vegetation productivity across the Sahel" uses gridded climate and vegetation data to determine the impact of seasonal rainfall metrics (typically ignored over large areas) on NPP. The analysis is performed over the Sahel where NPP estimates are used extensively for food security analysis and other important areas of drylands research. The manuscript is generally well written and organized. The analysis is thorough and sufficiently addresses the objectives of the manuscript. The discussion and summary adequately capture the major findings. I believe the manuscript should be accepted by Biogeosciences after the authors address the few questions/comments below

[Figure]

1) Regarding grammar: the sentences in the introduction and discussion tend to be long-winded, omit commas, and confuse "that" and "which." Sections, subsections, etc. should be numbered 1., 1.1, 1.1.1. throughout the manuscript. Use past tense for tasks performed and present tense for general statements. There are other minor grammatical and spelling errors that should be addressed.

2) The methods section would flow better if rainfall and NDVI were detailed in their own data subsection.

3) RFE-2.0 is no longer a "state of the art" dataset and is not appropriate for daily rainfall estimation. RFE-2.0 is primarily used at 10-day intervals. The developers caution against using the daily product, because the estimates are statistically disaggregated from the 10-day data and therefore may or may not represent the physical reality. Why was the RFE successor CHIRPS not used for the analysis? It is higher resolution and I would suspect provides more realistic daily rainfall estimates...Why was daily data necessary if it was compared alongside 8-day MODIS?

4) Regarding MOD09Q1...the 8-day composites are quite noisy over the Sahel due to persistent cloud cover. Was any filtering done prior to S-G? Was the optimized MODIS S-G used? If so, please provide citation. Otherwise, how did you determine the smoothing terms? Certainly not a requirement for this manuscript, but the authors should consider using eMODIS in the future, since it is a 10-day product intended to be analyzed alongside RFE-2.0 or CHIRPS for food security applications.

5) The relationships in Figure 5 are non-linear. Why were they not fit with an exponential curve? How do you take into account the non-linearity of NDVI in highly productive grid cells?

Minor

Ln 54-57: Sentence beginning with "Recent studies..." is difficult to understand and should be reworded.

Ln 100: Typo "(R. Fensholt and Rasmussen, 2011)"

Ln 117-130: Consider using a different nomenclature for climatological and dynamic rainfall anomalies.

Font sizes in the figures are too small.

---

## Referee Comment (RC2) · Anonymous Referee #2 · 16 Oct 2017

General comments

The well-written and well-structured manuscript presents an analysis about the impact of the seasonal rainfall distribution on ANPP in the Sahel zone during the period 2001-2015. The authors utilized a gridded dataset of daily precipitation to compute different seasonal rainfall metrics and related these to NDVI SIN (derived from a time series of MOD09Q1) as a proxy for ANPP. The objective of the manuscript is addressed with a sound methodology and the findings of the authors are supported by the results. Overall, the topic is very interesting and relevant, e.g. for the food security and climate change community, and I support the acceptance of the manuscript after minor revisions. In general, I would like the authors to address a few more issues in the discussion/conclusion. First, please discuss the quality of the utilized data products, es-

pecially the rainfall dataset, and if this could affect the obtained results. Second, please address the possibility of an adaptation of the vegetation (e.g. change in species composition) to a change in the seasonal rainfall distribution (related to the last paragraph in the summary and conclusion section). Furthermore, I would be interested if the authors tested if there is a relationship (correlation) between the different analysed seasonal rainfall metrics. But this does not need to be part of the paper (it is just curiosity). More specific comments are given below.

Specific comments

Line 12: "number of rainy days, rainfall intensity, number of consecutive dry days and heavy rainfall events" -> please specify that these metrics refer to the rainy season

Line 17: Please add a half sentence to shortly explain the meaning of "breakpoints" in this context

Line 26: remove "KM" from the reference

Line 29: "wet season which can be highly variable between years" -> the wet season is highly variable in time and space (please add the space component)

Line 34: "21st century climate change" -> add "predicted"

Line 89: Provide a reference/website where to access the RFE-2.0 data

Line 90: Please provide an explanation why you opted for the RFE-2.0 dataset and not another daily precipitation dataset like CHIRPS (see also comment of M. Marshall)

Line 109: "on day (Pi)" -> Is there something missing, e.g. "on a certain day"?

Line 122: Provide a reference for the MOD09Q1 product and/or a website where to access the product

Line 128: Please specify if you did the resampling of the NDVI data before or after applying TIMESAT

Line 129: Provide a reference for both land cover maps; Specify how the masking was done (i.e., did you mask out water if both LC maps indicate water in a pixel or if at least one of the LC indicate water?)

Line 133: Please explain why you chose the Pearson's correlation coefficient and not for example the non-parametric Spearman's rank correlation coefficient

Line 134: Please provide the R package name for the GAMs

Line 135: "Team, 2014" should be "R Team, 2014"

Line 138: "individual rainfall variables" -> please use terms consistently, e.g. use "individual seasonal rainfall metrics" here

Line 151f: "The 95th percentile of NDIV SIN [...] for a given rainfall amount" -> It is not fully clear how you calculated the potential vegetation productivity: Did you calculate it pixel-wise? Does the "given rainfall amount" represent the mean annual rainfall sum of a pixel? -> please clarify the description of your calculations

Line 158: "with a later onset" -> should it not be "with an earlier onset"?

Line 166f: "with a near linear relationship" -> Fig. 4 does not look like a linear relationship

Line 182: "variations in seasonal rainfall distribution" -> Do you mean all seasonal rainfall metrics?

Line 239: "from where the rather low amount of vegetation loses sensitivity to even more extreme seasonal distribution" -> please reformulate as the phrase is not really clear

Line 255: RESTREND approach -> please explain a bit more this approach

Line 367f: Some information like publisher are missing for this publication

Line 390f: Information about journal volume, issue and pages missing

[Figure]

Line 400: Some information like publisher are missing for this publication

Line 432f: Should be "R Team" instead of "Team, R"; there is twice the year 2014

Table 1: Definition of CDD: should it not include "during the wet season" or something similar

Figure 1: Here the study area is defined as the area between 100-700 mm annual rainfall. But in the description of the study area on page 4 you define your study area as the area between 100-800 mm annual rainfall -> Please choose one definition and use it consistently throughout the paper

Figure 5: Please provide a parameter and a unit for the color scale in the sub-figures

Figure 6: Maybe provide an r value for each sub-figure as in figure 5

---

## Author Response (AR1)

**Responses to comments:**

We thank the assocciate editor for these additional suggestions. As for the "However, I agree with anonymous referee 2 that the relationship presented in figure 4 is a non-linear relationship."
We have now removed the mentioning of linearity.

As for "Furthermore I suggest to present the "data availability" section not following "summary and conclusion", but in the method section."
We have moved that section accordingly.

**M. Marshall Referee #1**

The manuscript titled, "Impacts of the seasonal distribution of rainfall on vegetation productivity across the Sahel" uses gridded climate and vegetation data to determine the impact of seasonal rainfall metrics (typically ignored over large areas) on NPP. The analysis is performed over the Sahel where NPP estimates are used extensively for food security analysis and other important areas of drylands research. The manuscript is generally well written and organized. The analysis is thorough and sufficiently addresses the objectives of the manuscript. The discussion and summary adequately capture the major findings. I believe the manuscript should be accepted by Biogeosciences after the authors address the few questions/comments below

We would like to thank Dr. Marshall for the supportive comments on the manuscript. We have implemented all of the suggestions and we believe that the revised version is now substantially improved. Below we respond to each of the comment individually.

1) Regarding grammar: the sentences in the introduction and discussion tend to be long-winded, omit commas, and confuse "that" and "which." Sections, subsections, etc. should be numbered 1., 1.1, 1.1.1. throughout the manuscript. Use past tense for tasks performed and present tense for general statements. There are other minor grammatical and spelling errors that should be addressed.

R1: Thanks. We have edited the introduction and have rephrased some overly convoluted sentences. We have carefully gone through the manuscript to remove additional minor errors and the subsections have been numbered as suggested.

2) The methods section would flow better if rainfall and NDVI were detailed in their own data subsection.

R2: In the revised version we have chosen to present the CHIRPS rainfall data in a separate subsection 2.2 (L91), whereas the use of the MODIS NDVI is now included as a part of the section describing the TIMESAT seasonal integration approach "2.4 Estimation of growing season ANPP" (L123).

3) RFE-2.0 is no longer a "state of the art" dataset and is not appropriate for daily rainfall estimation. RFE-2.0 is primarily used at 10-day intervals. The developers caution against using the daily product, because the estimates are statistically disaggregated from the 10-day data and therefore may or may not represent the physical reality. Why was the RFE successor CHIRPS not used for the analysis? It is higher resolution and

I would suspect provides more realistic daily rainfall estimates: : :Why was daily data necessary if it was compared alongside 8-day MODIS?

R3: Thank you for suggesting the use of the CHIRPS data. We fully agree and the CHIRPS data was used to replace RFE in the revised MS.

4) Regarding MOD09Q1: : :the 8-day composites are quite noisy over the Sahel due to persistent cloud cover. Was any filtering done prior to S-G? Was the optimized

MODIS S-G used? If so, please provide citation. Otherwise, how did you determine the smoothing terms? Certainly not a requirement for this manuscript, but the authors should consider using eMODIS in the future, since it is a 10-day product intended to be analyzed alongside RFE-2.0 or CHIRPS for food security applications.

R4: We thank the reviewer for this good suggestion. Here, we applied a Savitsky-Golay filter on the 8-day MODIS composites. On the one hand, Savitsky-Golay filtering has been reported to be robust against noise and missing data, while preventing over-smoothing. On the other hand, as the presence of clouds is one major error sources of EO data in the Sahel region during the rainy season, we routinely checked the quality of MODIS time series via TIMESAT GUI, where one can test how does Gaussian/ Double Logistic/ Savitsky-Golay filter smooth the data and cope with noise removal. For this study (and some other studies published recently), the Savitskey-Golay filter was chosen since it can filter out the noise and capture the temporal features for the time series, whereas the other two filters generally over-smooth the temporal curve. As described in the revised version of the MS, we have set a window size of 4 and seasonal parameter of 0.5 to fit one season per year, number of iterations for upper envelope adaptation of 2, and strength of the envelope adaptation of 2.

The eMODIS is indeed a product identified more cloud observations in some regions such as Canada and US. It will definitely be considered in the future analysis along also with initial tests that we have conducted based on the MODIS MAIAC processing chain.

We added in the text as: "*For this study, we applied the Savitzky-Golay filter implemented in TIMESAT with the following settings: A window size of 4 was applied and a seasonal parameter of 0.5 to fit one season per year. Both the number of iterations for upper envelope adaptation and strength of the envelope adaptation were set to 2 and the start and end of season were determined as 20% and 50% of the amplitude respectively.*" (L130-134)

5) The relationships in Figure 5 are non-linear. Why were they not fit with an exponential curve? How do you take into account the non-linearity of NDVI in highly productive grid cells?

R5: Yes, thanks - they are not exactly linear relationships for the most variables. We used exponential regression in the revised version to better represent the statistical relationship between variables.

**Minor**

Ln 54-57: Sentence beginning with "Recent studies: : :" is difficult to understand and should be reworded.

R6: Thanks, we have rephrased the sentence. (L55)

Ln 100: Typo "(R. Fensholt and Rasmussen, 2011)"

R7:  Sorry for this. This part in this new version is deleted and other similar problems were checked throughout the manuscript.

Ln 117-130: Consider using a different nomenclature for climatological and dynamic rainfall anomalies.

R8: In order to keep consistent with the work by Liebmann et al., we have adopted their nomenclature.

Liebmann, B., Bladé, I., Kiladis, G. N., Carvalho, L. M. V, Senay, G. B., Allured, D., … Funk, C. (2012).

Seasonality of African precipitation from 1996 to 2009. Journal of Climate, 25(12), 4304–4322.

https://doi.org/10.1175/JCLI-D-11-00157.1

Font sizes in the figures are too small.

R9: Thanks, we have enlarged font size for the figures.

**Anonymous Referee #2**

General comments

The well-written and well-structured manuscript presents an analysis about the impact of the seasonal rainfall distribution on ANPP in the Sahel zone during the period 2001-2015. The authors utilized a gridded dataset of daily precipitation to compute different seasonal rainfall metrics and related these to NDVI SIN (derived from a time series of MOD09Q1) as a proxy for ANPP. The objective of the manuscript is addressed with a sound methodology and the findings of the authors are supported by the results. Overall, the topic is very interesting and relevant, e.g. for the food security and climate change community, and I support the acceptance of the manuscript after minor revisions. In general, I would like the authors to address a few more issues in the discussion/conclusion. First, please discuss the quality of the utilized data products, especially the rainfall dataset, and if this could affect the obtained results. Second, please address the possibility of an adaptation of the vegetation (e.g. change in species composition) to a change in the seasonal rainfall distribution (related to the last paragraph in the summary and conclusion section). Furthermore, I would be interested if the authors tested if there is a relationship (correlation) between the different analysed seasonal rainfall metrics. But this does not need to be part of the paper (it is just curiosity). More specific comments are given below.

We would like to thank the reviewer for the supportive comments and suggestions, which have certainly helped us in improving the manuscript. We have implemented all of the suggestions and we believe that the revised version of the MS is now substantially improved. Below we respond to each of the comment individually.

General comments

**First,** please discuss the quality of the utilized data products, especially the rainfall dataset, and if this could affect the obtained results.

R1: Thanks. We have adopted reviwer1's suggestion and your comment (L90) and the CHIRPS data were used to replace RFE, We discuss data quality in the beginning of the discussion section of the revised version: (L267-278) " *Uncertainty in the rainfall data is inevitably to have an impact on the extraction of seasonal rainfall metrics which further impacts the relationship between seasonal rainfall metrics and ANPP. Based on improved climatologies systematic bias in the CHIRPS dataset has been removed and the data is considered state-of-the-art within quasi-global, high spatial resolution rainfall datasets (Funk et al., 2015). As this study does not address temporal changes in the seasonal rainfall metrics or $\sum NDVI$, but merely presents results on the general coupling*

*between rainfall metrics and vegetation productivity, we consider the results to be statistically robust. We*
*conducted a parallel set of analyses based on the RFE-2.0 rainfall product developed by the NOAA Climate*
*Prediction Center (CPC) (Herman et al., 1997), which, like CHIRPS, is also a gauge-satellite blended and the*
*outcome of these analyses yielded nearly similar results as what is presented here. At the same time ∑NDVI derived*
*from MODIS will also be impacted from cloud cover during the growing season, but the use of the Savitzky-Golay*
*filtering algorithm has proven to be an effective way of overcoming residual noise effects in the NDVI time-series*
*(Fensholt et al., 2015)."*

**Second,** please address the possibility of an adaptation of the vegetation (e.g. change in species composition) to a
change in the seasonal rainfall distribution (related to the last paragraph in the summary and conclusion section).

R2: This is an excellent point, which is on our to-do-list, however it would require long-term records of field
observations of herbaceous species composition to study further – but that would certainly be interesting. We have
added the following sentence to this section (L353-356).

*"Inter-annual differences in the seasonal distribution of rainfall is known to have an impact on species*
*composition in Sahel (Mbow et al., 2013) and it is likely the herbaceous vegetation is able to adapt to changes*
*seasonal rainfall distribution expressed by a shift in the abundance of species favored by increased heavy rainfall*
*events and longer dry spells."*

**Furthermore,** I would be interested if the authors tested if there is a relationship (correlation) between the
different analysed seasonal rainfall metrics. But this does not need to be part of the paper (it is just curiosity).

R3: The non-parametric Spearman's rank correlation coefficients between all seasonal rainfall metrics are
showed in the below:

|           | R     | Onset | Cessation | RD    | SDII  | R95sum | CDD |
|-----------|-------|-------|-----------|-------|-------|--------|-----|
| R         | 1     |       |           |       |       |        |     |
| Onset     | -0.79 | 1     |           |       |       |        |     |
| Cessation | 0.60  | -0.45 | 1         |       |       |        |     |
| RD        | 0.92  | -0.83 | 0.60      | 1     |       |        |     |
| SDII      | 0.69  | -0.37 | 0.32      | 0.38  | 1     |        |     |
| R95sum    | -0.91 | 0.82  | -0.60     | -0.99 | -0.35 | 1      |     |
| CDD       | -0.82 | 0.67  | -0.33     | -0.86 | -0.41 | 0.84   | 1   |

**Specific comments**

Line 12: "number of rainy days, rainfall intensity, number of consecutive dry days and heavy rainfall events" -> please specify that these metrics refer to the rainy season

R4: We changed the text as 'This study tests the importance of rainfall metrics in the wet season (onset and cessation of the wet season, number of rainy days, rainfall intensity, number of consecutive dry days and heavy rainfall events) on growing season ANPP' . (L11)

Line 17: Please add a half sentence to shortly explain the meaning of "breakpoints" in this context

R5: Thanks, we have included the following sentence in the abstract: "We analyzed critical breakpoints for all metrics to test if vegetation response to changes in a given rainfall metric surpasses a threshold beyond which vegetation functioning is significantly altered." (L17-18)

Line 26: remove "KM" from the reference

R6: Sorry for this. It has been changed accordingly. (L28)

Line 29: "wet season which can be highly variable between years" -> the wet season is highly variable in time and space (please add the space component)

R7: Thanks. It has been changed accordingly. (L31)

Line 34: "21st century climate change" -> add "predicted"

R8: Thanks, we have changed it as suggested.

Line 89: Provide a reference/website where to access the RFE-2.0 data

R9: RFE-2 was replaced with CHIRPS, so this paragraph has been deleted. We added the website and reference for CHIRPS.

Line 90: Please provide an explanation why you opted for the RFE-2.0 dataset and not another daily precipitation dataset like CHIRPS (see also comment of M. Marshall)

R10: Since we fully agree that CHIRPS data will be a better choice than RFE, we have followed this suggestion and replaced all the analyses by CHIRPS data.

Line 109: "on day (Pi)" -> Is there something missing, e.g. "on a certain day"?

R11: Thanks. We rephrased this sentence to make it clearer.

Line 122: Provide a reference for the MOD09Q1 product and/or a website where to access the product

R12: Thanks. We have added a "Data availability" section following "Summary and conclusion". (L358)

Line 128: Please specify if you did the resampling of the NDVI data before or after applying TIMESAT

R13: Thanks. We have rephrased this part a bit to make it clearer. We first calculated the $\sum$NDVI and then the
resampling was done. e.g., The derived small seasonal integral was used as the $\sum$NDVI. The method is well
established and proven to be a reliable proxy for the growing season ANPP in Sahel (Olsson et al., 2005; Rasmus
et al., 2013). The $\sum$NDVI data was then aggregated to the resolution of CHIRPS (0.05 ˚) using a bilinear resampling
method. (L134)

Line 129: Provide a reference for both land cover maps; Specify how the masking was
done (i.e., did you mask out water if both LC maps indicate water in a pixel or if at least
one of the LC indicate water?)

R14: Thanks for your suggestion. It has now been specified that pixels with water were masked out if one or
both of the land cover products indicated the presence of water in a pixel. (L136-137)

Line 133: Please explain why you chose the Pearson0s correlation coefficient and not
for example the non-parametric Spearman's rank correlation coefficient

R15: We thank the reviewer for this comment. We have used the non-parametric Spearman's rank correlation
coefficient in the new version since it measures the monotonic relationship which is a better choice in this study.
This has now been corrected in the text.  (L141)

Line 134: Please provide the R package name for the GAMs

R16:  Thanks. The MGCV package is provided.

Line 135: "Team, 2014" should be "R Team, 2014"

R17: Thanks. It has been corrected accordingly.

Line 138: "individual rainfall variables" -> please use terms consistently, e.g. use "individual
seasonal rainfall metrics" here

R18: Thanks for your suggestion; it has been done accordingly throughout the MS.

Line 151f: "The 95th percentile of NDIV SIN [: : :] for a given rainfall amount" -> It is not
fully clear how you calculated the potential vegetation productivity: Did you calculate it
pixel-wise? Does the "given rainfall amount" represent the mean annual rainfall sum of
a pixel? -> please clarify the description of your calculations

R19: Thanks, we have rephrased this sentence (L162-166). We added '*The 95th percentile of $\sum$NDVI was*
*selected to represent the potential vegetation productivity attainable for a given seasonal rainfall metric (Donohue*
*et al., 2013). Seasonal rainfall metrics were binned according to the dynamic range of the individual metrics and*
*the average 95th percentile of $\sum$NDVI was calculated for each bin (for onset, cessation and RD bins with an interval*

*of 1 were used; for SDII a bin of 0.3 was applied; for R95sum bin intervals were set to 0.02; finally we used bins*

*0.5 for CDD)'.*

Line 158: "with a later onset" -> should it not be "with an earlier onset"?

R20: Sorry for this mistake and it has been changed to 'earlier'.

Line 166f: "with a near linear relationship" -> Fig. 4 does not look like a linear relationship

R21: Here we simply want to show the spread of ∑NDVI for each rainfall zone and when considering the different quantiles of the plot, we actually believe that it's correct to present this Fig. 4 relationship as near-linear.

Line 182: "variations in seasonal rainfall distribution" -> Do you mean all seasonal rainfall metrics?

R22: Yes, the term seasonal rainfall distribution was used to indicate seasonal rainfall metrics. To make it clearer, we have now added a few examples "(e.g., onset and R95sum)". (L198)

Line 239: "from where the rather low amount of vegetation loses sensitivity to even more extreme seasonal distribution" -> please reformulate as the phrase is not really clear

R23: Sorry for this. We have rephrased this sentence to: "from where the vegetation loses sensitivity to the impact from an increased frequency of heavy rainfall events" (L258-259)

Line 255: RESTREND approach -> please explain a bit more this approach

R24: Thanks, we have added this part a parenthesis explaining the approach: "regressing $\sum$NDVI from annual precipitation and subsequently calculating the residuals as the difference between observed $\sum$NDVI and $\sum$NDVI

as predicted from annual precipitation)" (L285-289)

Line 367f: Some information like publisher are missing for this publication

R25: Sorry for this mistake. It was an IPCC report and we specified this in the L417.

Line 390f: Information about journal volume, issue and pages missing

R26: Sorry for this. The missing information was added. (L441)

Line 400: Some information like publisher are missing for this publication

R27: Added. (L451)

Line 432f: Should be "R Team" instead of "Team, R"; there is twice the year 2014

R28: Sorry for the mistake. Corrected. (L479)

Table 1: Definition of CDD: should it not include "during the wet season" or something

Similar

R29: Thanks. We rephrased the sentence, which now reads: "Maximum number of consecutive days with
rainfall <1 mm during wet season"
Figure 1: Here the study area is defined as the area between 100-700 mm annual
rainfall. But in the description of the study area on page 4 you define your study area
as the area between 100-800 mm annual rainfall -> Please choose one definition and
use it consistently throughout the paper
R30: Sorry for this mistake. We used 100-800 mm $yr^{-1}$ and it has been changed throughout the MS accordingly.
Figure 5: Please provide a parameter and a unit for the color scale in the sub-figures
R31: Thank, we have added 'Density' as the title for the color scale.
Figure 6: Maybe provide an r value for each sub-figure as in figure 5
R32: Since there are three variables shown in each sub-figure, we have decided to report the r values in a separate
analysis (Figure 7).

[revised manuscript text omitted]

**2.3 Deriving rainfall seasonality metrics**

The method used to identify the onset and cessation of the wet season is referred to Liebmann et al. (2012)

and applicable to multiple datasets for precipitation seasonality analysis across the African continent (Dunning et al., 2016). As is described in Liebmann et al. (2012), the climatology wet season is initially determined by the climatological cumulative daily rainfall anomaly, $A(d)$, –calculated from the long-term (2001-2015) average rainfall $(R_i)$ for each day of the calendar year minus the long-term annual-mean daily average $(\bar{R})$. The day with minimum value $(d_s)$ is defined as the start of the wet season and the maximum point $(d_c)$ marks the end of the wet season.

(1)

$$A(d) = \sum_{i=1\,Jan}^{d} R_i - \bar{R}$$

Subsequently, the onset and cessation were calculated individually for each year and each grid point. For each year the extraction of the seasonality of the wet season was based on equation (2). The daily cumulative rainfall anomaly $A(D)$ on a certain day $(P_i)$ was computed for each day in the range $d_s$ -50 to $d_c$ +50 for each year and the day with minimum value is considered as the onset of the wet season.

(2)

$$A(D) = \sum_{j=d_s-50}^{D} P_i - \bar{R}$$

Once the onset and cessation dates of the wet season for each year were found, the remaining variables were calculated (Table 1). Fig. 2a illustrates an example of daily rainfall for the grid point (13.5˚ N, 5.0˚ W) in 2001 and the corresponding cumulative daily anomaly curves are shown in Fig 2b. The blue and red lines signify $A(d)$ and

$A(D)$, respectively. The range of minimum and maximum points in the blue line denotes the climatological wet season (Liebmann et al., 2012). The wet season of each individual year was then determined based on the daily precipitation observations covered by the climatological wet season. Areas where the annual minimum occurs after the 1st of October (desert areas) were excluded from further analysis (Diaconescu et al., 2015).

**2.4 Estimation of growing season ANPP**

The growing season integrated  NDVI ($\sum$NDVI)SIN was used as a proxy for the growing season ANPP. The

$\sum$NDVI was derived using TIMESAT (Jönsson and Eklundh, 2004), a widely used tool to extract vegetation seasonal metrics. For this study, we applied the Savitzky-Golay filter and determined the start and end of season at

20% and 50% of the amplitude respectively. The method is well established and proven to be a reliable proxy for the growing season ANPP in Sahel (Olsson et al., 2005; Rasmus et al., 2013). The $\sum$NDVI data was then aggregated to the resolution of CHIRPSRFE 2.0 (11×11 km0.05 ˚) using a bilinear resampling method. Both Globeland30

(Chen et al., 2014) and ESA CCI (2010) land cover maps (https://www.esa-landcover-cci.org/) were used to mask water bodies, irrigated and flooded areas if one of the both land cover products indicates the XX.

**2.5 Statistical analyses**

[revised manuscript text omitted]

st+ed.+1995&oq=.+Woody+Plants+in+Agro-Ecosystems+of+Semi-

Arid+Regions%3A+with+an+Emphasis+on+the+Sahelian+Countries%2C+Softcover+reprint+of+the+original+1

st+ed.+1995&aqs=chrome..69i57.392j0j4&sourceid=chrome&ie=UTF-8 (Accessed 24 July 2017), 1995.

Chen, J., Chen, J., Liao, A., Cao, X., Chen, L., Chen, X., He, C., Han, G., Peng, S., Lu, M., Zhang, W., Tong,

X. and Mills, J.: Global land cover mapping at 30 m resolution: A POK-based operational approach, ISPRS

Journal of Photogrammetry and Remote Sensing, 103, 7–27, doi:10.1016/j.isprsjprs.2014.09.002, 2014.

Diaconescu, E. P., Gachon, P., Scinocca, J. and Laprise, R.: Evaluation of daily precipitation statistics and monsoon onset/retreat over western Sahel in multiple data sets, Climate Dynamics, 45(5–6), 1325–1354, doi:10.1007/s00382-014-2383-2, 2015.

Diouf, A. A., Hiernaux, P., Brandt, M., Faye, G., Djaby, B., Diop, M. B., Ndione, J. A. and Tychon, B.: Do agrometeorological data improve optical satellite-based estimations of the herbaceous yield in Sahelian semi-arid ecosystems?, Remote Sensing, 8(8), 668, doi:10.3390/rs8080668, 2016.

Donohue, R. J., Roderick, M. L., McVicar, T. R. and Farquhar, G. D.: Impact of CO2 fertilization on maximum foliage cover across the globe's warm, arid environments, Geophysical Research Letters, 40(12),

3031–3035, doi:10.1002/grl.50563, 2013.

Dunning, C. M., Black, E. C. L. and Allan, R. P.: The onset and cessation of seasonal rainfall over Africa,

Journal of Geophysical Research: Atmospheres, 121, 11,405-11,424, doi:10.1002/2016JD025428, 2016.

Evans, J. and Geerken, R.: Discrimination between climate and human-induced dryland degradation, Journal of Arid Environments, 57(4), 535–554, doi:10.1016/S0140-1963(03)00121-6, 2004.

Fay, P. A., Carlisle, J. D., Knapp, A. K., Blair, J. M. and Collins, S. L.: Altering rainfall timing and quantity in a mesic grassland ecosystem: Design and performance of rainfall manipulation shelters, Ecosystems, 3(3), 308–

319, doi:10.1007/s100210000028, 2000.

Fensholt, R. and Proud, S. R.: Evaluation of earth observation based global long term vegetation trends —

Comparing GIMMS and MODIS global NDVI time series, Remote Sensing of Environment, 119, 131–147, doi:10.1016/j.rse.2011.12.015, 2012.

Field, C.: Managing the risks of extreme events and disasters to advance climate change adaptation, in IPCC

special report of the intergovernmental panel on climate change. [online] Available from:

https://www.google.com/books?hl=en&lr=&id=nQg3SJtkOGwC&oi=fnd&pg=PR4&dq=Managing+the+risks+o f+extreme+events+and+disasters+to+advance+climate+change+adaptation:+special+report+of+the+intergovern mental+panel+on+climate+change&ots=12LhuvrDOM&sig=GmCG5J4SpH4 (Accessed 4 June 2017), 2012.

Fischer, E. M., Beyerle, U. and Knutti, R.: Robust spatially aggregated projections of climate extremes,

Nature Climate Change, 3(12), 1033–1038, doi:10.1038/nclimate2051, 2013.

Fitzpatrick, R. G. J., Bain, C. L., Knippertz, P., Marsham, J. H. and Parker, D. J.: The West African monsoon onset: A concise comparison of definitions, Journal of Climate, 28(22), 8673–8694, doi:10.1175/JCLI-D-15-

0265.1, 2015.

Funk, C., Peterson, P., Landsfeld, M., Pedreros, D., Verdin, J., Shukla, S., Husak, G., Rowland, J., Harrison, L., Hoell, A. and Michaelsen, J.: The climate hazards infrared precipitation with stations—a new environmental record for monitoring extremes, Scientific Data, 2, 150066, doi:10.1038/sdata.2015.66, 2015.

Guan, K., Good, S. P., Caylor, K. K., Sato, H., Wood, E. F. and Li, H.: Continental-scale impacts of intra-seasonal rainfall variability on simulated ecosystem responses in Africa, Biogeosciences, 11(23), 6939–6954, doi:10.5194/bg-11-6939-2014, 2014.

Guan, K., Sultan, B., Biasutti, M., Baron, C. and Lobell, D. B.: What aspects of future rainfall changes matter for crop yields in West Africa?, Geophysical Research Letters, 42(19), 8001–8010, doi:10.1002/2015GL063877, 2015.

Herrmann, S. M., Anyamba, A. and Tucker, C. J.: Recent trends in vegetation dynamics in the African Sahel and their relationship to climate, Global Environmental Change, 15(4), 394–404, doi:10.1016/j.gloenvcha.2005.08.004, 2005.

Houerou, H. Le: Rain use efficiency: a unifying concept in arid-land ecology, Journal of Arid Environments, 7(3), 213–247 [online] Available from: http://cat.inist.fr/?aModele=afficheN&cpsidt=9676430 (Accessed 14 July 2017), 1984.

Huber, S., Fensholt, R. and Rasmussen, K.: Water availability as the driver of vegetation dynamics in the African Sahel from 1982 to 2007, Global and Planetary Change, 76(3–4), 186–195, doi:10.1016/j.gloplacha.2011.01.006, 2011.

Jönsson, P. and Eklundh, L.: TIMESAT—a program for analyzing time-series of satellite sensor data, Computers & Geosciences, 30, 833–845 [online] Available from: http://www.sciencedirect.com/science/article/pii/S0098300404000974 (Accessed 4 June 2017), 2004.

Kaspersen, P. S., Fensholt, R. and Huber, S.: A spatiotemporal analysis of climatic drivers for observed changes in Sahelian vegetation productivity (1982–2007), International Journal of Geophysics, 2011, 1–14, doi:10.1155/2011/715321, 2011.

Kharin, V. V., Zwiers, F. W., Zhang, X. and Hegerl, G. C.: Changes in temperature and precipitation extremes in the IPCC ensemble of global coupled model simulations, Journal of Climate, 20(8), 1419–1444, doi:10.1175/JCLI4066.1, 2007.

Lebel, T. and Ali, A.: Recent trends in the Central and Western Sahel rainfall regime (1990–2007), Journal of

Hydrology, 375(1), 52–64, doi:10.1016/j.jhydrol.2008.11.030, 2009.

Leisinger, Schmitt, K.: Survival in the Sahel: An ecological and developmental challenge., 1995.

Liebmann, B., Bladé, I., Kiladis, G. N., Carvalho, L. M. V, Senay, G. B., Allured, D., Leroux, S. and Funk,

C.: Seasonality of African precipitation from 1996 to 2009, Journal of Climate, 25(12), 4304–4322, doi:10.1175/JCLI-D-11-00157.1, 2012.

Muggeo, V. M. R.: Estimating regression models with unknown break-points, Statistics in Medicine, 22(19),

3055–3071, doi:10.1002/sim.1545, 2003.

Nicholson, S. E.: The nature of rainfall variability over Africa on time scales of decades to millenia, Global and Planetary Change, 26(1–3), 137–158, doi:http://dx.doi.org/10.1016/S0921-8181(00)00040-0, 2000.

Olsson, L., Eklundh, L. and Ardö, J.: A recent greening of the Sahel—trends, patterns and potential causes,

Journal of Arid Environments, 63(3), 556–566, doi:10.1016/j.jaridenv.2005.03.008, 2005.

Panthou, G., Vischel, T. and Lebel, T.: Recent trends in the regime of extreme rainfall in the Central Sahel,

International Journal of Climatology, 34(15), 3998–4006, doi:10.1002/joc.3984, 2014.

Prince, S., Colstoun, D. and Brown, E.: Evidence from rain-use efficiencies does not indicate extensive

Sahelian desertification, Global Change Biology, 4(4), 359–374, doi:10.1046/j.1365-2486.1998.00158.x, 1998.

Rasmus, F. and Rasmussen, K.: Analysis of trends in the Sahelian "rain-use efficiency" using GIMMS NDVI,

RFE and GPCP rainfall data, Remote Sensing of Environment, 115(2), 438–451, doi:10.1016/j.rse.2010.09.014,

2011.

Rasmus, F., Rasmussen, K., Kaspersen, P., Huber, S., Horion, S. and Swinnen, E.: Assessing land degradation/recovery in the African Sahel from long-term earth observation based primary productivity and precipitation relationships, Remote Sensing, 5(2), 664–686, doi:10.3390/rs5020664, 2013.

Ratzmann, G., Gangkofner, U., Tietjen, B. and Fensholt, R.: Dryland vegetation functional response to altered rainfall amounts and variability derived from satellite time series data, Remote Sensing, 8, 1026, doi:10.3390/rs8121026, 2016.

Rishmawi, K., Prince, S. and Xue, Y.: Vegetation Responses to Climate Variability in the Northern Arid to

Sub-Humid Zones of Sub-Saharan Africa, Remote Sensing, 8(11), 910, doi:10.3390/rs8110910, 2016.

Romankiewicz, C., Doevenspeck, M., Brandt, M. and Samimi, C.: Adaptation as by-product: Migration and environmental change in Nguith, Senegal, Journal of the Geographical Society of Berlin, 147(2), 95–108, doi:10.12854/erde-147-7, 2016.

Sanogo, S., Fink, A. H., Omotosho, J. A., Ba, A., Redl, R. and Ermert, V.: Spatio-temporal characteristics of the recent rainfall recovery in West Africa, International Journal of Climatology, 35(15), 4589–4605, doi:10.1002/joc.4309, 2015.

Simon Richard, P. and Rasmussen, L. V.: The influence of seasonal rainfall upon Sahel vegetation, Remote

Sensing Letters, 2(3), 241–249, doi:10.1080/01431161.2010.515268, 2011.

Smith, M. D.: The ecological role of climate extremes: Current understanding and future prospects, Journal of

Ecology, 99(3), 651–655, doi:10.1111/j.1365-2745.2011.01833.x, 2011.

Taylor, C. M., Belušić, D., Guichard, F., Parker, D. J., Vischel, T., Bock, O., Harris, P. P., Janicot, S., Klein,

C. and Panthou, G.: Frequency of extreme Sahelian storms tripled since 1982 in satellite observations, Nature,

544(7651), 475–478, doi:10.1038/nature22069, 2017.

R. Team,: R: A language and environment for statistical computing. Vienna, Austria: R Foundation for

Statistical Computing, http://www.R-project.org/., 2014.

Thomey, M. L., Collins, S. L., Vargas, R., Johnson, J. E., Brown, R. F., Natvig, D. O. and Friggens, M. T.:

Effect of precipitation variability on net primary production and soil respiration in a Chihuahuan Desert grassland, Global change biology, 17(4), 1505–1515, doi:10.1111/j.1365-2486.2010.02363.x Effect, 2011.

Vries, F. P. de and Djitèye, M.: The productivity of Sahelian rangeland: a study of soils, vegetation and the exploitation of this natural resource., Pudoc. [online] Available from:

http://www.cabdirect.org/abstracts/19826740725.html (Accessed 17 August 2017), 1982.

Wessels, K., Prince, S., Malherbe, J. and Small, J.: Can human-induced land degradation be distinguished from the effects of rainfall variability? A case study in South Africa, Journal of Arid Environments, 68(2), 271–

297, doi:10.1016/j.jaridenv.2006.05.015, 2007.

Wood, S.: Generalized additive models: an introduction with R, CRC press. [online] Available from:

https://www.google.com/books?hl=en&lr=&id=JTkkDwAAQBAJ&oi=fnd&pg=PT18&dq=Wood,+S.N.+Genera lized+Additive+Models,+An+Introduction+with+R,+1st+ed.%3B+Chapman+%26+Hall/CRC+Texts+in+Statisti cal+Science&ots=7eHi1gLZOj&sig=PxSG6SZXNN1yJc1fbfFKyz7PH5M (Accessed 14 July 2017a), 2017.

[revised manuscript text omitted]